# Superlative mechanical energy absorbing efficiency discovered through self-driving lab-human partnership

Kelsey L. Snapp [1], Benjamin Verdier[2], Aldair E. Gongora [1], Samuel Silverman [2], Adedire D. Adesiji [1], Elise F. Morgan [1,3,4], Timothy J. Lawton [5], Emily Whiting[2] & Keith A. Brown [1,3,6] ✉

Energy absorbing efficiency is a key determinant of a structure's ability to provide mechanical protection and is defined by the amount of energy that can be absorbed prior to stresses increasing to a level that damages the system to be protected. Here, we explore the energy absorbing efficiency of additively manufactured polymer structures by using a self-driving lab (SDL) to perform >25,000 physical experiments on generalized cylindrical shells. We use a human-SDL collaborative approach where experiments are selected from over trillions of candidates in an 11-dimensional parameter space using Bayesian optimization and then automatically performed while the human team monitors progress to periodically modify aspects of the system. The result of this human-SDL campaign is the discovery of a structure with a 75.2% energy absorbing efficiency and a library of experimental data that reveals transferable principles for designing tough structures.

Structural motifs define the ways we efficiently use materials. For instance, the ubiquity of I-beams in architecture is due to the efficiency of this shape in resisting both shear and bending[1,2]. Natural structures feature similar examples, such as the hollow circular cross-section of bamboo providing high bending and torsional resistance[3–6]. For the large class of structures designed to provide protection under a compressive load, the key property to consider is the total mechanical energy absorbed during compression[7–10]. This desire to discover tough structures has motivated a focus on metrics like energy dissipated per unit volume or per unit weight. However, in compression, it is nearly always possible to increase the applied stress to absorb more energy, thus when considering specific energy absorbed, or another metric with a dimension, one must also define an operating stress. Along these lines, there are practical restrictions to absorbing energy in any engineering application, for example, that the stress must be held below a level that would damage the system to be protected[11]. Collectively, these restrictions mean that terms like specific energy

absorption are not easily applied in comparing tough structures at different stresses. Therefore, it is useful to define an energy-absorbing efficiency $K_s$, a non-dimensional measure of how much energy is absorbed without surpassing a given threshold stress[12,13]. Unfortunately, $K_s$ is difficult to optimize because most of the energy absorbed by a structure designed for mechanical protection occurs beyond the elastic regime where deformations are highly non-linear, often feature dynamic self-self contacts, and are challenging to model.

As a result of the challenge of designing tough structures, much work has focused on known, relatively simple motifs such as honeycomb lattices or cylindrical shells that have an analytical basis for performing well[12,14]. Others have drawn inspiration from nature to identify more complex structural motifs[15–18]. Computational approaches, including finite element analysis (FEA) and machine learning-based approaches have also been widely used to design complex geometries[19–25]. These computational approaches pair well with additive manufacturing, which allows the fabrication of complex

[1]Department of Mechanical Engineering, Boston University, Boston, MA, USA. [2]Department of Computer Science, Boston University, Boston, MA, USA. [3]Division of Materials Science & Engineering, Boston University, Boston, MA, USA. [4]Department of Biomedical Engineering, Boston University, Boston, MA, USA. [5]Soldier Protection Directorate, US Army Combat Capabilities Development Command Soldier Center, Natick, MA, USA. [6]Physics Department, Boston University, Boston, MA, USA. ✉e-mail: brownka@bu.edu

designs[26–30]. Nevertheless, the fabrication of candidate structures is often the limiting step in the design process and is commonly limited to validating designs. Furthermore, despite the speed and versatility of computational approaches such as FEA, it is very challenging and sometimes impossible to accurately capture $K_s$ using computation because of the complex interplay of material plasticity, material non-linearities, structural non-linearities, and dynamic self-self contacts[31–33]. Furthermore, models studied by FEA often deviate from reality due to unavoidable processing-dependent defects and variability of the physically realizable structures. Thus, improvements to $K_s$ remain slow: to date, additively manufactured structures designed for energy absorption typically feature $K_s < 50\%$ (Figure S1). There exist better synthetic materials, the best being a plastic foam reported to have reached $K_s = 68.1\%$[34]. However, this record is surpassed by nature: Balsa wood has the highest previously achieved $K_s$, 71.8%, showing the value of millions of years of evolution[35]. It is clear that new approaches are needed if the performance envelope of this important property is to be improved.

Here, we utilize a self-driving lab (SDL) to test >25,000 additively manufactured structures in a large-scale data-driven campaign to discover tough structures with superlative $K_s$. SDLs are robotic research systems that select, perform, and analyze physical experiments without needing human intervention[36,37], and they have been productively employed in chemistry[38,39], materials science[40], mechanics[41], and microscopy[42,43]. Motivated by the observations that SDLs can progress toward user-chosen goals faster than either high-throughput experimentation[44] or tests chosen by subject matter experts[45–48], we hypothesize that an SDL allowed to explore seven polymers in an 11-dimensional parameter space over trillions of possible designs can discover new structural motifs that advance the frontier of $K_s$. The result of this sustained human-machine collaboration is that we realize a structure with $K_s = 75.2\%$. In addition to showing the opportunities for SDLs to overcome design barriers, this campaign results in a vast, labeled dataset that has implications for both mechanics and design more generally. For instance, we explore two high-performing structural motifs and find that they exhibit consistent performance within classes of materials, namely plastic or hyperelastic polymers. Finally, aggregate analysis of this data provides general design heuristics that allow for the efficient selection of materials and structures.

## Results

### Defining a Campaign to Study Generalized Cylindrical Shells

As a motivating example to explore the considerations that define and limit $K_s$, we consider the compressive behavior of a cylindrical shell composed of a hyperelastic thermoplastic polyurethane (TPU). When tested in compression, the resulting force $F$-displacement $D$ curve shows an initial elastic region, a yield point, and then complex post-yield behavior that originates from combinations of plastic deformation, buckling and other large elastic deformations, and reentrant contact (Fig. 1a). To compute $K_s$, $F$-$D$ is first converted to stress $\sigma$ vs. strain $\varepsilon$ for the effective medium using the dimensions of the component (Figure S2). Defining $K_s$ requires specifying a threshold stress $\sigma_t$ that is typically associated with the strength of the system to be protected. Graphically, $K_s$ represents the amount of energy absorbed by the component while $\sigma \leq \sigma_t$ (Fig. 1a – blue region) relative to the maximum energy that could be absorbed during complete compression ($\varepsilon = 1$) without exceeding $\sigma_t$ (Fig. 1a – red rectangle). To compute $K_s$ at a specific $\sigma_t$, we numerically evaluate $K_s = \sigma_t^{-1} \int_0^{\varepsilon_t} \sigma(\varepsilon) d\varepsilon$ where $\varepsilon_t$ is the greatest strain at which $\sigma \leq \sigma_t$ for all $0 < \varepsilon \leq \varepsilon_t$. Interestingly, for most structures, $K_s(\sigma_t)$ has a single well-defined maximum $K_s^*$ at an optimum threshold stress $\sigma_t^*$. This $\sigma_t^*$ often corresponds to the initial yield stress of a structure, although it can occur at higher stresses, particularly when the component densifies at low strains or does not significantly soften after yielding. As such, $K_s^*$ and $\sigma_t^*$ were found for this and all structures numerically as described in Section 5.6 of the Supplemental

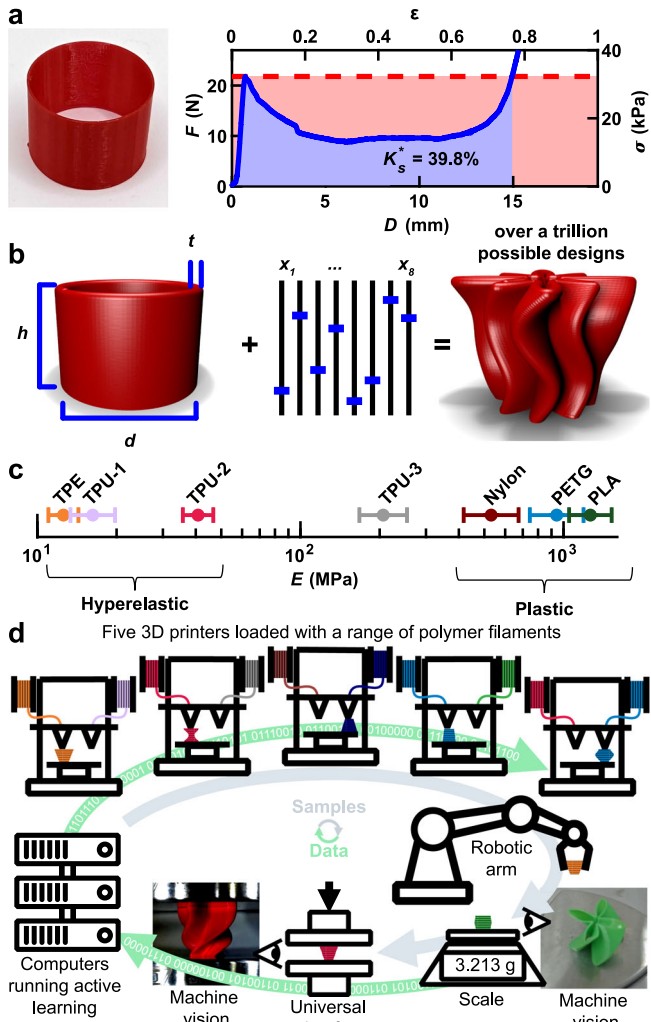

**Fig. 1 | Challenge of designing energy-absorbing structures. a** Force $F$ vs. displacement $D$ and effective medium stress $\sigma$ vs. compressive effective medium strain $\varepsilon$ for an additively manufactured cylindrical shell made of thermoplastic polyurethane (TPU). Maximum energy absorbing efficiency $K_s^*$ is calculated at an optimum threshold stress $\sigma_t^*$ (dashed line) by dividing the energy absorbed while $\sigma \leq \sigma_t^*$ (blue region) by the theoretical maximum amount absorbed (red rectangle). **b** Eleven independent geometric parameters, including diameter $d$, height $h$, wall thickness $t$, and eight other parameters $x_{1-8}$ that together define a generalized cylindrical shell (GCS). When combined, at least trillions of unique designs are possible. **c** Elastic modulus $E$ of the seven polymers studied as determined by compression tests. Error bars represent one standard deviation. **d** Schematic showing an autonomous research system in which five 3D printers are used to fabricate polymeric structures that are automatically weighed, imaged, and tested using quasistatic compression. The output of this testing is automatically interpreted and used to select subsequent designs to test.

Information. In the example of Fig. 1a, the cylindrical shell is limited to $K_s^* = 39.8\%$ due to significant post-yield softening. To maximize $K_s^*$, a flat post-yield region and a delay of densification until large $\varepsilon$ are both desirable. Unfortunately, this knowledge alone does not provide a prescription for how to adjust the structure to obtain these desired behaviors.

We hypothesized that programmed perturbations to the geometry of a cylindrical shell could tailor the complex post-yielding behavior to drastically increase $K_s^*$. While cylindrical shells are typically defined by a small number of geometric parameters, namely their diameter $d$, height $h$, and thickness $t$, we augmented these to form an 11-parameter family of structures termed generalized cylindrical shells (GCS) (Fig. 1b). In addition to $t$, $h$, and $d$, a GCS is defined by eight

additional parameters $x_{1-8}$ including four that adjust the cross-sections of the top and bottom of the shell, one to define the perimeter of the top relative to the perimeter of the bottom, and three to describe the rotation of the perturbations from top to bottom (Figure S3). Furthermore, because the GCS space does not have circular cross sections for most designs, we define $d = \frac{P}{2\pi}$, where $P$ is the average perimeter of the design. We note that each of the eleven parameters would only need twelve unique values for their combinations to surpass a trillion unique designs. Given that each parameter is continuous and can be assigned many more than twelve values, we consider trillions of unique designs to be a lower limit to the size of the parameter space. While we have previously used FEA to compute mechanical metrics like resilience, stiffness, and yield force[49], our inability to rapidly or accurately calculate the shape of the post-yield region using FEA led us to not pursue this method for accelerating the study of $K_s$. Therefore, experiments were necessary for assessing $K_s$. Because all the resulting structures are topologically equivalent to cylindrical shells, they can be fabricated using extrusion-based additive manufacturing by continuously extruding material, thus making this parameter space intrinsically designed for additive manufacturing. In addition to the geometric parameters, we sought to explore a variety of polymers. Therefore, we considered seven materials that included those that are hyperelastic, such as a thermoplastic elastomer (TPE) and TPU, those that plastically deform, such as polylactic acid (PLA), and materials that fall in between these two distinct classes (Fig. 1c). These materials can be characterized based on their elastic modulus $E$, their plateau strength $\sigma_p$, and rebound strain (Figure S4).

To efficiently search the effectively infinite GCS parameter space, we employed the Bayesian experimental autonomous researcher (BEAR), a customized SDL developed to combine additive manufacturing of polymers and mechanical testing (Fig. 1d). The BEAR is a closed-loop system in which samples are printed using one of five fused filament fabrication (FFF) printers, automatically retrieved using a six-axis robotic arm, and then characterized using a scale, machine vision, and uniaxial compression testing. After testing, the information was automatically analyzed to determine whether the test was of acceptable quality. This learning process featured a fault tolerance that was a combination of intrinsic and explicit factors. Especially if components had no structure due to large overhangs, the small contact area with the print bed, non-manifold surfaces due to high twist values, or print-head collisions caused by excessive extrusion, they would be tested and exhibit low efficiency, naturally guiding the algorithm away from this region of design space. If the sample was so poorly defined that it could not even be tested, these experiments were automatically marked as unprintable to prevent the algorithm from repeatedly selecting this or similar designs. In both cases, these assignments were manually confirmed asynchronously by the human team. Subsequent experiments were selected using Bayesian optimization, which entailed conditioning a surrogate model of the mechanical performance using all previously measured GCS components and then selecting combinations of designs and materials that maximized a specified acquisition function. As this experimental campaign began with a very small amount of data, Bayesian optimization using Gaussian process regression was selected to model this data because it is efficient in the low-data regime[50]. Now that a large database exists, more sophisticated models such as variation autoencoders could be employed to more accurately model the design space and facilitate in the selection of new experiments[51]. The SDL autonomously performed these tasks to choose, perform, and analyze experiments at a typical pace of ~50 experiments per day. Collectively, 25,387 experiments were performed using seven different materials, with 13,250 experiments resulting in valid data. It is worth emphasizing that even experiments not included in the final dataset provide value, for example, in determining the subspace of GCS designs that are printable using FFF. From the beginning of the campaign to the end,

experiments were running for ~60% of the total wall clock time, showing the robustness of this process. This system is an evolution of an SDL developed by our research group[44]. A picture of the system (Figure S5) as well as full details on the hardware (Figure S7) and software (Figure S8) used as part of the BEAR are provided in the methods and supplementary information.

## Discovering high-performing structures

An extensive SDL campaign proceeded as a continuous human-machine collaboration where the responsibilities were shared between the SDL and the human team (Figure S9). Progress in the campaign can be visualized by tracking $K_s^*$ measured for each experiment along with a running maximum throughout the campaign (Fig. 2a). The continuous progression was a result of both persistent experimentation by the SDL and choices made by the human team based on the progress of the SDL. Interestingly, large jumps in performance were typically either due to serendipity (*i.e.* the SDL chose a fortuitous region) or a human-led intervention. For example, we highlight three human interventions based on observing the progress of the SDL. First, prior to experiment 4,829, the SDL was programmed to select experiments based on $K_s$ at one specific $\sigma_t$. However, we noted that there were several different reasons why a specific sample would have a low $K_s$, so we needed to provide the SDL with more information. We hypothesized that tracking both $K_s^*$ and $\sigma_t^*$ from each experiment would allow for more meaningful information to be extracted by the SDL. After implementing this change, the SDL rapidly increased the maximum $K_s^*$ from 45% to 55%. As a second example, at experiment 15,678, we noted that a large fraction of plastic components were failing the height quality control check but passing the mass check. We had been heating the print bed after printing to facilitate the automated removal of components, but determined that the forces exerted during removal could deform plastic components. Upon changing the SDL to cool plastic components prior to removal, the system proceeded to make a series of jumps in maximum $K_s^*$ from 60 to 68%. Finally, at experiment 17,730 we noted that the predictive model used by the SDL was systematically underpredicting $K_s^*$ for high-performing components, so we implemented a process where the proposed experiment was selected using a model built only on data close to the best-observed experiment, a process similar to algorithms such as TURBO or ZOMBI[52,53]. This intervention led the SDL to progress from 70.6% to 75.2% in maximum $K_s^*$. A summary of significant human-led actions is provided in Figure S10.

The culmination of these adjustments and continued experimentation by the SDL resulted in the observation of $K_s^* = 75.2\%$, a value that was higher than had been previously reported. The performance of this superlative experiment is shown in Fig. 2b, which shows the $\sigma - \varepsilon$ curve and photographs of the component at different stages of compression. It is clear from the flatness of the post-yield region, together with the photographs, that the SDL has discovered a way for buckling and other large elastic deformations, plasticity, and reentrant contact to work together to achieve a remarkably flat plateau until densification initiated at ~80% strain. Interestingly, this component was composed of PLA, which is not commonly regarded as a high-performance material. Upon repeated experiments, the design, which we termed Palm, had an average $K_s^* = 73.1 \pm 0.9$. Although Palm printed in PLA had the largest single value of $K_s^*$, 75.2%, observed in the entire campaign, we discovered other components that had higher average $K_s^*$ values than Palm.

## Material influence on design and performance

To explore variations in performance across different material classes, we selected two high-performing designs discovered in different materials. The design discovered in PLA with the highest average $K_s^*$, termed Willow, is tall and has a compact center region (Fig. 3a). Testing 15 identically prepared samples of the Willow design resulted in a tight

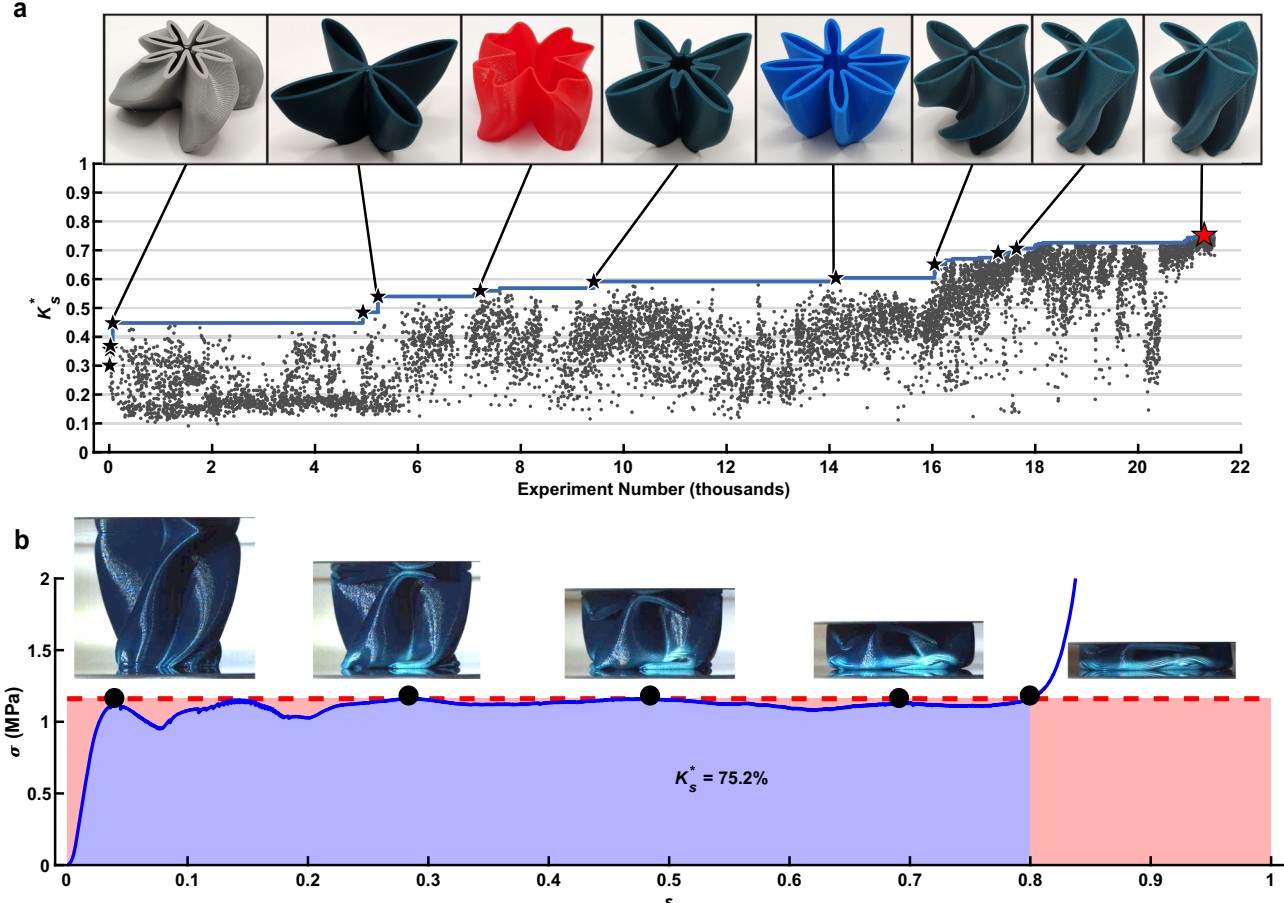

**Fig. 2 | Research campaign to find highly efficient structures. a** Each maximum energy absorbing efficiency $K_s^*$ measured over the first ~21,500 experiments performed. Pictures highlight noteworthy components (black stars) and the highest-performing structure (red star). Larger versions of images are included as Figure S11. The color of the pictured components is indicative of the material used, with Green indicating PLA, Blue indicating PETG, and Red/Gray indicating different types of

TPU. The solid blue line denotes the running best $K_s^*$ observed. **b** Effective medium stress $\sigma$ vs. effective medium strain $\varepsilon$ for experiment 21,285, named Palm, which resulted in $K_s^* = 75.2\%$. Inset photographs show the state of the component at various points indicated on the curve (images enhanced to improve clarity – originals given as Figure S12). Shading denotes regions used to compute $K_s^*$ as described in Fig. 1a.

distribution of yield forces with variations in the post-yield plateau. Nevertheless, PLA components made using this design exhibit $K_s^* = 73.3 \pm 0.9\%$, showing a consistent performance above previously reported maxima. That said, the repeatability of superlative designs on multiple FFF systems suggests that defects in the fabrication process do not limit performance. The highest performing design discovered for TPU-2 is termed Iroko and consists of a relatively short and open design (Fig. 3b). We observed more substantial variations among the 15 Iroko $\sigma - \varepsilon$ profiles, and the average $K_s^* = 53 \pm 4\%$ was substantially lower than that of the best plastic components. The differences in attainable $K_s^*$ between PLA and TPU-2 can be explained by considering that these are different material classes, with PLA being a glassy polymer that exhibits substantial plastic deformation while TPU-2 is a hyperelastic elastomer. This difference in properties is most evident in their post-compression behavior, in which the TPU component recovers ~99% of its height one minute after compression while the PLA component is permanently flattened to ~23% of its initial height.

While Willow and Iroko represent optimizations for their original materials, the question remains of whether the performance of these shapes can translate to other materials or if it is a highly bespoke optimization of this combination of material and design. To explore this, components based on the Willow and Iroko designs were fabricated using a wide range of materials and tested in triplicate (Figure S13). Studying $K_s^*$ of these components showed the limitations of

the transferability of these designs (Fig. 3c). While each design performed with comparable $K_s^*$ for materials in their respective classes (i.e. hyperelastic vs. plastic), a transition region was observed at intermediate $E$. This observation reveals how material stiffness and plasticity modulate the energy-absorbing capacity of geometric designs. Specifically, higher stiffness, together with greater plasticity, mitigates the amount of softening the component exhibits as portions of it bend during compression. Overall, the comparison of Willow and Iroko confirmed that designs perform well within specific classes of materials but that these geometric motifs do not directly translate to different material classes.

## Broader design considerations

While the SDL-based campaign was able to discover highly efficient designs, we hypothesized that the broader corpus of mechanical tests performed during this campaign could provide further mechanical insight. As an initial exploration of this idea, the results of all the experiments performed with TPU-2 are shown in Fig. 4a. The shaded region denotes the convex hull that estimates the space of accessible properties. This shows that the best performance observed for this material occurs at a single $\sigma_t^*$, which we denote the material peak threshold stress $\sigma_{tp}$. Interestingly, all other materials studied exhibit a similarly shaped convex hull with a distinct peak (Figure S14), highlighting both the importance and the feasibility of tuning the material

properties to the specific energy-absorbing application. We found that over the seven tested materials, $\sigma_{tp}$ was strongly correlated with the polymer plateau stress $\sigma_p$ (Fig. 4b), providing an algorithmic process for selecting a material to optimally match use cases across a wide range of threshold stresses determined from different systems to be protected. Then the material can be selected and structured to maximize $K_s$ at that $\sigma_t$. Importantly, we do not believe that $\sigma_{tp}$ is dependent upon 3D

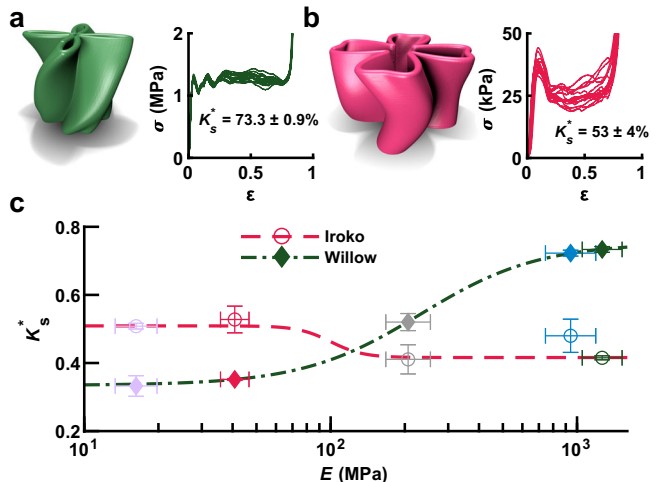

**Fig. 3 | Exploration of high-performing designs discovered in elastic and plastic materials. a** Rendering of Willow, a high-performing design discovered using the plastic polymer polylactic acid (PLA) together with effective medium stress $\sigma$ vs. effective medium strain $\varepsilon$ for 15 identically prepared PLA Willow components. **b** Rendering of Iroko, a high-performing design discovered using the hyperelastic polymer TPU-2, together with $\sigma$ vs. $\varepsilon$ for 15 identically prepared TPU-2 Iroko components. **c** Measured maximum energy absorbing efficiency $K_s^*$ vs. polymer elastic modulus $E$ for Iroko and Willow components made from one of five polymers. Dashed lines show a sigmoidal fit to guide the eye. Error bars represent one standard deviation. Marker colors denote component composition, as shown in Fig. 1c.

printing capabilities or choice of experiments as this value was not found to drift to higher or lower stresses throughout the campaign. As such, this relationship may have applicability beyond the GCS design family.

In addition to tuning material properties, we hypothesized that unifying features of high-performing designs could be extracted to provide transferable guidance for realizing efficient structures. For example, it is reasonable to expect that the relative density $\rho_r$ of the component would strongly influence $K_s^*$ (Fig. 4c). This hypothesis is motivated by the observation that the two factors that together bound $K_s^*$ are the flatness of the plateau region and the strain $\varepsilon_d$ where this plateau drastically rises due to densification. It has been observed for foams that $\varepsilon_d$ is bounded by relative density[12,34]. Because $\varepsilon_d \geq K_s^*$, we hypothesized that low $\rho_r$ is necessary for high $K_s^*$. Examining $K_s^*$ vs. $\rho_r$, we found that $K_s^*$ peaked at $\rho_r \sim 0.1$ with all designs with $K_s^* \geq 65\%$ having $0.05 \leq \rho_r \leq 0.21$, providing guidance for structural design. Interestingly, because $\rho_r$ can be calculated prior to fabrication, limiting physical testing to designs with $\rho_r$ in this range can eliminate potential components that are unable to achieve high $K_s^*$. To leverage this fact, we implemented a decision-making policy that focused on designs in this low-density regime. This is both an example of using a metric that is quick to compute to accelerate learning and represents a facet of the human-machine teaming where an observation by the human team helped the SDL search more efficiently.

Beyond the aggregate details of the design, there is a great deal of work exploring the mechanical regimes present for cylindrical shells under uniaxial compression. For example, the ratio $d/t$ of a cylindrical shell determines whether plastic cylinders fail through plastic deformation (thick wall limit) or fail elastically through the formation of local buckles (thin wall limit)[54]. This transition has been observed to occur at $d/t \sim 100$. Further, the height of cylindrical shells is often characterized by the dimensionless length parameter $\omega = h/\sqrt{dt}$[55]. Here, cylinders are considered to be short when $\omega < 1.7$. Plotting $K_s^*$ vs. $d/t$ and $\omega$ reveals that all of the highest performing structures (*i.e.* $K_s^* \geq 70\%$), which were made from plastic materials, had $16 < d/t < 24$ and $6.75 < \omega < 8.25$, which can be considered thick-walled medium-length cylindrical shells (Fig. 4d). For simple cylinders in this region, one would expect elastic

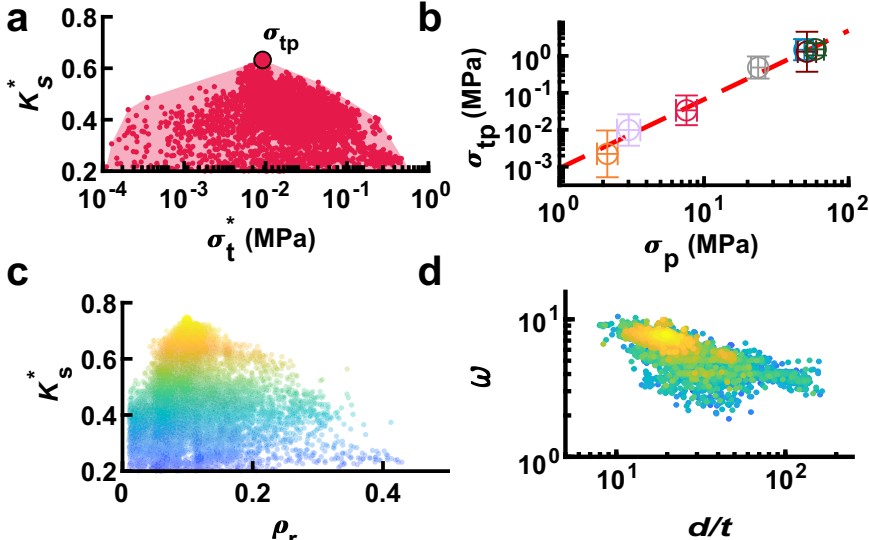

**Fig. 4 | Design insights that emerge from mechanical dataset. a** Experimental data for TPU-2 with each dot representing maximum energy absorbing efficiency $K_s^*$ vs. optimum threshold stress $\sigma_t^*$ for a specific experiment and the shaded region denoting the convex hull of the entire performance space corresponding to that material. The maximum point, termed material peak threshold stress $\sigma_{tp}$, is the $\sigma_t^*$ at which the highest $K_s^*$ is observed for that material. **b** $\sigma_{tp}$ vs. polymer plateau stress $\sigma_p$ for seven polymers tested during the campaign together with a power law fit shown as a dashed line. Error bars indicate one standard deviation of $\sigma_{tp}$ found throughout the campaign. Marker color indicates material as shown in Fig. 1c. **c** $K_s^*$ vs. relative density $\rho_r$ for all components tested during the campaign with point color denoting $K_s^*$. **d** Normalized height $\omega$ vs. diameter-to-thickness ratio $d/t$ for all components tested during the campaign in which point color denotes $K_s^*$ as in (**c**).

buckling and plastic deformation to both play major roles. Thus, one way to understand the data-driven optimization process is that the other eight geometric parameters that define a GCS component have been tuned to guide these complex buckling and plastic interactions to interact constructively. Interestingly, we may use the tools of machine learning to identify which geometric motifs are most responsible for this improvement. In particular, we employ Shapley additive explanations (SHAP) analysis to find that the four-lobed profile of the cylinder ($x_{2-3}$) together with the linear and sinusoidal twist of this profile along the shell ($x_{6-8}$) are together responsible for 90% of the improvement over a simple cylindrical shell (Figure S15). Mechanically, this suggests that the key feature for improving the efficiency is producing local plastic deformation events that result in sufficient self-self contacts to prevent post-yield weakening. This feature allows the structure to maximize the material plasticity that occurs while maintaining a flat post-yield region.

While the main focus of this work has been the discovery of isolated structures that exhibit high mechanical efficiency and, therefore, high toughness, it is interesting to consider whether these components could act in concert to be useful for larger systems. For instance, these structures could be tiled in two dimensions to form a regular lattice. To facilitate this, the minimum tileable area of each component was used as the reference area to compute stress. To explore whether the aggregate $K_s$ of multiple high-performing components would retain their high performance, we computed the predicted $K_s$ values for combinations of the Willow replicates reported as part of Fig. 3a. Interestingly, not only was the composite predicted to retain its superlative $K_s$, but this value was predicted to increase as more components were added to a maximum of 74.2% when 10 components were combined (Figure S16). This improvement with averaging can be understood by considering that the largest variability observed for this component was in the plateau region and so regression of this curve to its mean is expected to result in closer agreement between the initial yield stress and this plateau region. These results point to examining combinations of GCS components as a promising area of future research.

This study focused on optimizing the maximum energy absorbing efficiency, but it is also interesting to consider the performance of these components through the lens of other commonly used metrics such as energy absorption normalized by mass or volume (Figure S17). Despite this not being the optimization target, structures that absorbed as much as 22.8 J/g or 10 J/cm³ were identified as part of this study.

## Discussion

This work reports a series of mechanical insights that resulted from performing an extensive experimental campaign using an SDL. Through the exploration of a vast parameter space, we were able to identify components with superlative $K_s$, advancing the frontier of energy absorption and finally overcoming the record previously held by nature. These designs were found to be general within material classes when additively manufactured, showing the opportunities and limitations of transferability of the designs. A longstanding goal in the mechanics community is to identify simple shapes that are predicted to obtain high performance, such as the analytically predicted 75% efficiency of honeycomb lattices[12]. However, a noteworthy feature of this work is that the final components contain repeatable geometric details defined by the FFF process that are not captured in the simple GCS digital model, such as the scalloping of the walls. This acknowledges a key reality that, in practice, all structures have to be physically realized and any manufacturing technique contains characteristic microstructures and idiosyncrasies that are unique to that method. It may be that obtaining superlative efficiency requires optimizing the structure within the context of how it would be physically realized by the fabrication method – as we have done here through our extensive experimental campaign. We expect future work will focus on studying the transferability of these high performing GCS geometries to other fabrication methods.

In addition to identifying superlative designs, this SDL campaign also illustrated how optimization and learning could be complementary goals in that the generated corpus of data allowed for the extraction of design insights for the optimal use of polymers. These insights include matching polymer materials to target use cases, highlighting the use of $\rho_r$ as an aggregate descriptor, and gaining connections to the broader literature on the mechanics of cylindrical shells. While here we only present limited mechanical trends that capture the breadth of the dataset, we anticipate that the availability of this data will motivate others to dive deeper into specific regions and provide insight into phenomena such as failure mechanisms and what geometric aspects would prevent them. The assembled database of GCS properties under uniaxial compression could also be very valuable in accelerating the discovery of other categories of mechanical response through the principles of transfer learning. For example, we have previously shown that quasi-static performance can be used to predict impact performance in a data-driven manner[56]. Further, transfer learning using feature transformation can be used to develop output spaces in which correlations are easier to learn, as we have shown using FEA calculations[49]. As such, the GCS dataset is likely to be useful to accelerate learning other categories of mechanical response. From a mechanics standpoint, we expect that this work will provide geometric motifs that lead to more efficient and safer protective equipment.

From a broader learning perspective, this work shows how the iterative and collaborative interaction between SDLs and human teams can lead to sustained progress. We should note that this study was not dedicated to benchmarking the acceleration inherent to using SDLs. Prior work, including our own[44,49], has focused on such benchmarking, and ongoing research is focused on developing algorithms and processes to efficiently select experiments[53,57]. In this work, our main focus was discovering new mechanical structures, and we believe that this type of sustained campaign is an example of how SDLs can fruitfully exist in the materials discovery pipeline. An important point to consider is that the process used here allowed the human-SDL partnership to leverage the strengths of each member. In particular, it has been widely seen that algorithmic decision making is more efficient at navigating high-dimensional spaces than even expert selection of experiments[45-48]. The approach used here allowed the SDL to autonomously and continuously explore this space while the humans provided periodic guidance in the form of making adjustments to the algorithm or accessible space. Having the human out of the loop allowed the autonomous system to continuously progress while requiring human decision making at each point would have made the pace of experimentation impossible. While it would have been useful from a benchmarking perspective to construct a predetermined process for humans to interact with the SDL within a narrow scope, this rigid framework would not allow for the effective use of human creativity to adapt to unexpected problems. The fruitful partnership of humans and SDL raises the question of what general lessons or approaches can be applied in these cases. Some studies have emerged that begin to address the concept[58,59], and it is a promising area for further study. Finally, an important avenue for future research is learning how to effectively combine simulations such as FEA with experimental campaigns such as this to increase the rate of learning while reducing the burden of physical experimentation.

## Methods
### Design of generalized cylindrical shells
To provide a vast space of potential designs that are topologically identical to cylindrical shells, we developed a generalized cylindrical shell (GCS) family of structures, which is an 11-dimensional parameter space (Figure S3). Three of these parameters are common to any cylindrical shell, namely the shell height $h$, wall thickness $t$, and average

perimeter $P_O$. Beyond these three variables, eight additional parameters $\mathbf{x} = [x_1, x_2, x_3, x_4, x_5, x_6, x_7, x_8]$ are introduced that change the height-dependent cross-section of the shell. The azimuthally-dependent radius $r(z,\phi)$ is shifted by adding two cosine functions with set periodicities as inspired by the summed cosine design of mechanical structures[60]. In particular, $r$ at any given height $z$ and azimuthal angle $\phi$ is given by

$$r(z,\phi) = r_0(z)\big[1 + C_4(z)\cos\big(4(\phi + \phi_0(z))\big) + C_8(z)\cos\big(8(\phi + \phi_0(z))\big)\big], \tag{1}$$

Where $C_4(z)$ and $C_8(z)$ are amplitude prefactors to the summed cosines, $\phi_0(z)$ is a rotational offset, and $r_0(z)$ is a prefactor adjusted to set the height-dependent perimeter $P(z)$ of the shell. Each of these functions is defined by terms of $\mathbf{x}$. Specifically, we define,

$$P(z) = P_0\left[1 + x_1\left(\frac{z}{h} - \frac{1}{2}\right)\right], \tag{2}$$

such that $x_1$ is the difference between the perimeter at the top of the component and the perimeter at the bottom. The Python function scipy.optimize.minimize was used to minimize the error between the $P(z)$ and the actual perimeter of Eq. (1), estimated using Simpson's rule, by adjusting $r_0(z)$ at each layer.

Each summed cosine term is defined by specifying its value at the top and bottom of the shell and linearly interpolating between these points, specifically,

$$C_4(z) = x_2\frac{h-z}{h} + x_3\frac{z}{h}, \tag{3}$$

and

$$C_8(z) = x_4\frac{h-z}{h} + x_5\frac{z}{h}. \tag{4}$$

To determine the azimuthal offset of each layer, we include two ways in which this can vary with height, namely a linear and sinusoidal shift. Specifically, we define,

$$\phi_0(z) = x_6\frac{z}{h} + x_7\sin(2\pi x_8 z). \tag{5}$$

Code to generate standard triangle language (STL) models based on the GCS family of shapes is available https://github.com/bu-shapelab/gcs.

## Sample preparation
To fabricate samples, STL files were converted to G-code using Slic3r v.1.3.0. Filament rolls for 3D printing were purchased and used as received. They include three different types of thermoplastic polyurethane (TPU): TPU-1 (NinjaFlex-Ninjatek), TPU-2 (Cheetah-Ninjatek), and TPU-3 (Armadillo-Ninjatek). Additionally, four more filaments were used: thermoplastic elastomer (TPE) (Chinchilla-Ninjatek), nylon (Matterhackers Pro Series), polyethylene terephthalate glycol (PETG) (Matterhackers Pro Series), and polylactic acid (PLA) (eSun PLA+ and MakerGear). Samples were fabricated using MakerGear M3 printers with either a 0.5 or a 0.75 mm nozzle at 80 °C bed temperature, 250 °C nozzle temperature (except for PLA, which was printed using 220 °C), and 15 mm/s print speed using vase mode. The cylindrical shell sample in Fig. 1a was fabricated out of TPU-2 (Cheetah – Ninjatek) to be 19.5 mm tall, 27.9 mm wide, and have 0.5 mm thick walls.

## Mechanical characterization of samples
Quasi-static uniaxial compression was performed using an Instron 5965 with a 5 kN load cell at 2 mm/min until the force reached 4.5 kN or until the platens were separated by less than 0.4 mm. The resulting force-displacement data was converted to stress-strain by dividing the force by the area of a hexagon that would enclose the component and by dividing the displacement by the initial height, respectively (Figure S2). The mechanical energy absorbing efficiency $K_s$ vs. threshold stress $\sigma_t$ was computed by dividing the amount of energy absorbed below $\sigma_t$ by the maximum amount that could be absorbed without exceeding that stress.

To determine the mechanical properties of each roll of filament, solid cylinders (100% infill) were printed, measuring 16 mm tall and 8 mm in diameter. These cylinders were then tested in uniaxial compression at 2 mm/min. Force-displacement curves were converted to stress-strain curves by dividing the force by the component cross-sectional area and by dividing the displacement by the height, respectively. From the resulting stress-strain curves, three material properties were calculated: the modulus of the material, plateau stress $\sigma_p$, and the rebound fraction. The modulus $E$ was calculated by fitting a series of lines in windows of 0.05–0.25 strain and an initial strain location of 0 to 0.25 strain (to avoid toe regions), both in increments of 0.05. The largest slope observed was taken as the modulus for the sample. The $\sigma_p$ was the stress value at 25% strain. The rebound fraction was the height of the part after a one-minute relaxation period divided by the initial height before testing, both measured by the Instron. One cylinder was tested for each roll of filament used.

## Development of the Bayesian experimental autonomous researcher
In order to study the mechanical energy absorbed by additively manufactured components under uniaxial compression, we developed and utilized a self-driving lab (SDL). This system incorporated several distinct instruments, computers, and algorithms that worked in concert to select experiments, construct physical samples, and test them with minimal human intervention. From a hardware perspective, this system consisted of five fused filament fabrication (FFF) printers (MakerGear M3-ID) arrayed in an arc. In the center of this arc was a 6-axis robot arm (Universal Robotics UR5e). Also in the working radius of this arm was a scale (Sartorius CP225D) and a universal testing machine (UTM) (Instron 5965). The arm had a webcam (Logitech C930e) to allow for monitoring the flow of experiments, and there was a video camera (PixelLINK PL-D722) with a lens (Infinity InfiniMite Alpha) mounted facing the bottom platen of the UTM to record videos of the tests. A picture of the SDL is shown in Figure S5. The hardware and software organization of this system is shown in Figure S7.

A flow chart describing the core actions of this system is shown in Figure S8. At the most basic level, the system comprised a loop implemented in MATLAB 2021a (Mathworks Inc) in which the system repeatedly iterated through six potential actions, namely: (1) Use the accumulated data to select the design and material to be tested next given the available printer and materials. (2) Generate the digital design files needed to run the available printer. (3) Send the G-code file to the printer and begin printing the component. (4) Retrieve the completed component from the printer and weigh it using the scale. (5) Retrieve the component from the scale, place it on the platen of the universal testing machine, run the mechanical testing program, and then clear the component from the platen. (6) Read the results of the mechanical testing and update the accumulated data. The order of priority was tuned to maximize the throughput of the system by giving priority to actions that were likely to become bottlenecks. The details of these steps are given in the supplementary information.

## The research campaign
Over the course of the campaign, 25,387 experiments were performed (Figure S9). Although individual experiments were rarely selected by

hand, a variety of decisions were made by the researchers along the way (Figure S10). Changes were made to the parameter space under consideration, such as adding sinusoidal twist or switching to Latin hypercube sampling (LHS) (Figure S10a). Additionally, new materials were added to the campaign, and the mix of filaments loaded into the printers was adjusted to focus on specific goals (Figure S10b). Finally, various decision policies were used, including maximum variance, expected improvement, and upper confidence bound (Figure S10c).

## Data availability
The data generated in this study have been deposited in the OpenBU database under the accession code https://hdl.handle.net/2144/46687.

## Code availability
The code used to generate GCS designs is available at Github https://github.com/bu-shapelab/gcs and is archived in Zenodo with the identifier https://doi.org/10.5281/zenodo.10933596[61]. The code used to operate the SDL is available at Github https://github.com/KelseyEng/BEAR_ADTS and is archived in Zenodo with the identifier https://doi.org/10.5281/zenodo.10928452[62].

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

## Acknowledgements

This work was supported by Google LLC, the Boston University Rafik B. Hariri Institute for Computing and Computational Science & Engineering (2017-10-005), the National Science Foundation (CMMI-1661412), and the US Army CCDC Soldier Center (contract W911QY2020002). This work was funded by Honeywell Federal Manufacturing and Technologies through contract number N000471618. Honeywell Federal Manufacturing and Technologies, LLC operates the Kansas City National Security Campus for the United States Department of Energy/National Nuclear Security Administration under contract number DE-NA0002839.

## Author contributions

Kelsey L. Snapp: Conceptualization, Methodology, Software, Investigation, Writing-Original Draft. Benjamin Verdier: Conceptualization, Methodology, Software, Writing-Review & Editing. Aldair Gongora: Conceptualization, Methodology, Software, Writing-Review & Editing. Samuel Silverman: Software, Visualization, Writing-Review & Editing. Adedire D. Adesiji: Investigation, Writing-Review & Editing. Elise F. Morgan: Formal Analysis, Writing-Review & Editing. Timothy J. Lawton: Conceptualization, Methodology, Writing-Review & Editing. Emily Whiting: Conceptualization, Methodology, Writing-Review & Editing, Supervision. Keith A. Brown: Conceptualization, Methodology, Writing-Original Draft, Writing-Review & Editing, Supervision.

## Competing interests

The authors declare no competing interests.
