## [Peer Review File · Nature Communications]

Superlative Mechanical Energy Absorbing Efficiency
Discovered through Self-Driving Lab-Human PartnershipEditorial Note: This manuscript has been previously reviewed at another journal that is not operating a transparent peer review scheme. This document only contains reviewer comments and rebuttal letters for versions considered at Nature Communications. Mentions of the other journal have been redacted.

REVIEWER COMMENTS

Reviewer #2 (Remarks to the Author):

The authors have addressed my comments. I recommend acceptance of the manuscript.

Reviewer #4 (Remarks to the Author):

The authors have improved the quality of the paper presentation and discussion, based on the reviewers' comments.

I am still concerned about the sustainability of their research approach and find the comparison they made regarding the footprint of the experiment vs. FE analysis, very speculative!

I suggest they update the paper justifying why they ruled out FE analysis. It would have surely saved experimental effort at least for a quick screening.

For the rest, I agree that this is an interesting study and worth publication in Nature Communications.

Reviewer #5 (Remarks to the Author):

While the authors have addressed some of my concerns, I am still hesitant to accept this article for publication in Nature Communications. The main problem being the reproducibility of the employed methods. An important aspect of the work that led to improved results for the optimisation depends on the interaction between humans and the experimental/optimization approach. At the moment, it is impossible to reproduce these specific conditions as many of the decisions that have been made during the experiments were ad hoc, and without a predetermined strategy. While, as also the authors mention, I can understand that involvement of humans and algorithms can lead to beneficial results, it should be reproducible and effectively executed. This is not the case for the present study.

This leaves the main result of the article to be on the identification of new tough structures. However, here I have strong doubts about the generality of these results, and how applicable these motives can be employed in other applications. What are the main conclusions that can be drawn by analysing these motives?

REVIEWER COMMENTS

Reviewer comments are italicized
Our comments are in plain type
Edits to the manuscript are in blue

Reviewer #2 (Remarks to the Author):

The authors have addressed my comments. I recommend acceptance of the manuscript.

Reviewer #4 (Remarks to the Author):

The authors have improved the quality of the paper presentation and discussion, based on the reviewers' comments.

I am still concerned about the sustainability of their research approach and find the comparison they made regarding the footprint of the experiment vs. FE analysis, very speculative!

I suggest they update the paper justifying why they ruled out FE analysis. It would have surely saved experimental effort at least for a quick screening.

For the rest, I agree that this is an interesting study and worth publication in Nature Communications.

Our reason for not using FEA prior to our experimental campaign was that calculating accurate estimates of energy absorbing efficiency using FEA was beyond our capabilities because of the interplay between material plasticity, material non-linearities, structural non-linearities, and dynamic self-self contacts. Beyond these, there are necessarily deviations between the digital model and the final structure and our goal was studying physically realized structures. In the prior round of revisions, we included a passage in the conclusion calling for more research about how to fruitfully include FEA in future similar work. To address this Reviewer's concern, we have augmented this with an earlier passage spelling out more clearly why we did not use FEA:

While we have previously used FEA to compute mechanical metrics like resilience, stiffness, and yield force,⁴⁹ our inability to rapidly or accurately calculate the shape of the post-yield region using FEA led us to not pursue this method for accelerating the study of K_s . Therefore, experiments were necessary for assessing K_s .

In addition, we have edited the earlier discussion about FEA to both highlight virtues of FEA and clarify the language about the challenge inherent to using FEA for this task. This passage now reads:

And furthermore, despite the speed and versatility of computational approaches such as FEA, it is very challenging and sometimes impossible to accurately capture K_s using computation because of the complex interplay of material plasticity, material non-linearities, structural non-linearities, and dynamic self-self contacts.³¹⁻³³ Furthermore, models studied by FEA often deviate from reality due to unavoidable processing-dependent defects and variability of the physically realizable structures.

Reviewer #5 (Remarks to the Author):

While the authors have addressed some of my concerns, I am still hesitant to accept this article for publication in Nature Communications. The main problem being the reproducibility of the employed methods. An important aspect of the work that led to improved results for the optimisation depends on the interaction between humans and the experimental/optimization approach. At the moment, it is impossible to reproduce these specific conditions as many of the decisions that have been made during the experiments were ad hoc, and without a predetermined strategy. While, as also the authors mention, I can understand

that involvement of humans and algorithms can lead to beneficial results, it should be reproducible and effectively executed. This is not the case for the present study.

While any given experiment is reproducible within reasonable experimental error, the exact order of experiments would never be the same even if this study was performed without any human intervention. This is due to both experimental variations and inherent randomness in the algorithms. It seems that the main distinction with the present work is that we are providing details about all experiments and all decisions that were made during the experimental campaign. In a conventional study, most of these experiments would simply be discarded and the only data that would be reported would be the final optimized structure. We also believe that formulating a rigid predetermined strategy for running a campaign would mitigate the primary benefit of having human involvement in that it would prevent human creativity and ingenuity from addressing unexpected complications as they arise. In fact, we feel that the flexible human-machine collaboration in our work was a feature of the system rather than a liability. We have added a passage to the conclusion reflecting this disconnect.

While it would have been useful from a benchmarking perspective to construct a predetermined process for the humans to interact with the SDL within a narrow scope, this rigid framework would not allow for the effective use of human creativity to adapt to unexpected problems.

In seeking to modify the manuscript to further address these concerns, we have noted that the use of the term autonomous is perhaps misleading when it refers to the whole campaign as this was an inherent interaction between the humans and the self-driving lab. Thus, we have removed the word autonomous from the caption of Figure 2 and added a sentence to the abstract to reflect this interaction.

This campaign proceeded as a collaboration between the automated system and human team with each experiment selected in the high-dimensional parameter space using Bayesian optimization while the human team periodically modified aspects of the system in response campaign progress.

Finally, we have revised the title to better reflect this consideration.

Superlative Mechanical Energy Absorbing Efficiency Discovered through Self-Driving Lab-Human Partnership

This leaves the main result of the article to be on the identification of new tough structures. However, here I have strong doubts about the generality of these results, and how applicable these motives can be employed in other applications. What are the main conclusions that can be drawn by analysing these motives?

We have noted four areas of impact in mechanics. These are highlighted throughout the text and in the conclusion:

- 1) The discovery of the most mechanically efficient structure to date.
- 2) Noting that superlative structures are thick-walled medium-length cylinders with substructure that makes plasticity and buckling function cooperatively.
- 3) The observation that the superlative structures remain high performing when constructed out of similar materials.
- 4) A general scaling relationship that connects the ideal working stress with material strength.

REVIEWERS' COMMENTS

Reviewer #5 (Remarks to the Author):

In reviewing the response to my article I am not convinced by the answers of the authors, as they have not addressed comments I have on reproducibility of the study. My concerns were not that the system was not autonomous, I absolutely think that having human interventions can help the optimization process. Yet, if there is no protocol on when users should intervene in the study, and how they should do this, I do not believe the methodology can be reproduced or applied directly to other studies. This limits the impact of this study.